# Does food insecurity cause anxiety and depression? Evidence from the changing cost of living study

**Melissa Bateson**[1], **Coralie Chevallier**[2], **Elliott A. Johnson**[3], **Matthew T. Johnson**[3], **Kate E. Pickett**[4], **Daniel Nettle**[2,3]*

1 Biosciences Institute, Newcastle University, Newcastle upon Tyne, United Kingdom, 2 Département d'études cognitives, Ecole Normale Supérieure, Institut Jean Nicod, Université PSL, EHESS, CNRS, Paris, France, 3 Department of Social Work, Education and Community Wellbeing, Northumbria University, Newcastle upon Tyne, United Kingdom, 4 Department of Health Sciences, University of York, York, United Kingdom

* daniel.nettle@northumbria.ac.uk

## Abstract

Food insecurity is associated with increased odds of anxiety and depression, but it is unclear whether this effect is causal, and if so, the timescale over which it occurs. We conducted a preregistered analysis of an intensive longitudinal dataset, the Changing Cost of Living Study, to explore evidence for causal processes linking food insecurity to anxiety and depression. Data were collected monthly between September 2022 and August 2023 from panels of adults in the UK (n = 244) and France (n = 240). Food insecurity predicted higher concurrent symptoms of anxiety and depression (measured respectively with GAD-7 and PHQ-8), controlling for gender, age, time and mental health in the previous month. Effect sizes were similar for GAD-7 and PHQ-8 scores. Changes in food insecurity produced changes in symptoms (within-individual standardised effects: 0.15 [95% CI 0.08-0.22] on GAD-7 and 0.11 [95% CI 0.03-0.18] on PHQ-8). The deterioration in mental health when participants became food insecure was of similar magnitude to the improvement observed when they became food secure. Consistent with Granger causality, food insecurity in the current month predicted poorer mental health in the following month after controlling for current mental health. These results support the hypothesis that food insecurity causes symptoms of anxiety and depression. The effects were rapid, occurring within a month of becoming food insecure, and were equally rapidly reversed. We conclude that policy interventions designed to reduce food insecurity would have immediate, clinically relevant, positive impacts on mental health.

## Introduction

There is growing evidence for the role played by social determinants in the development of poor mental health [1]. In the current study we focus on food insecurity, which

**Data availability statement:** The raw data and analysis code are available at https://osf.io/3rby2/.

**Funding:** The following funding was received: Agence Nationale de la Recherche, Grant/Award Numbers: ANR-21-CE28-0009 (CC) and ANR-23-CE28-0005-01 (DN); UK Prevention Research Partnership, Grant/Award Number: MR/S037527/1 (KEP); National Institute for Health and Care Research, Grant/Award Number: NIHR154451 (MJ); University of York Cost of Living Research Group (KEP) The funders played no role in the design of the study, the analysis, or the decision to publish.

**Competing interests:** The authors declare that they have no competing interests.

is defined as, 'the limited or uncertain availability of nutritionally adequate and safe foods or the limited or uncertain ability to acquire acceptable foods in socially acceptable ways' [2]. Food insecurity is associated with increased odds of stress, anxiety and depression [3–5]. However, the explanation for these associations is currently unclear. Identifying the mechanisms involved could inform the design of interventions to improve mental health. A barrier to formulating and testing mechanistic hypotheses is that most published data come from cross-sectional, observational studies that do not allow reliable inferences about causality. In the current study, we used longitudinal data to explore whether fluctuations in food-insecurity status within individuals covaried with short-term changes in symptoms of anxiety and depression, as would be predicted if there is a direct causal relationship between food insecurity and mental health.

Causal relationships between food insecurity and mental health could occur in both directions [6]. A range of (non-mutually exclusive) mechanisms by which food insecurity causes poor mental health has been proposed. For example, the poor diets associated with food insecurity could cause inadequate intake of nutrients necessary for mental health; uncertainty over the ability to acquire sufficient food could induce chronic psychological and physiological stress; and the pressure to acquire food in socially unacceptable ways could induce negative feelings of frustration, alienation, shame and guilt [7–10]. While such effects of food insecurity are generally assumed to be pathological, an alternative account assumes that food insecurity triggers fatigue and apathy as adaptive psychological mechanisms to reduce energy expenditure when access to food is limited [11]. The reverse causal pathway, whereby poor mental health causes food insecurity through its impacts on an individual's ability to work or access food, has also been discussed [6]. It is plausible that causal pathways operate in both directions and that food insecurity and poor mental health are mutually reinforcing [8].

In the absence of experiments manipulating food insecurity, observational datasets provide the main source of evidence available to test causal pathways. Establishing causality requires that three necessary conditions are met [12]. If experience of food insecurity causes a decline in mental health then: (i), there must be a statistically reliable empirical association between food insecurity and poorer mental health; (ii), this association must not be explained by other causes; and (iii), an increase in food insecurity must temporally precede a decline in mental health.

Consistent with the first condition, cross-sectional studies provide robust evidence for correlations between food insecurity and various indicators of poor mental health. Recent meta-analyses of this literature show that food insecurity is associated with higher odds of both depression and anxiety, with larger effects reported for depression than anxiety [3,5,13]. Support for the second condition is provided by evidence that these associations are present across all global regions [13], that they are independent of potentially-confounding socioeconomic factors and that the associations are dose-dependent, with more severe food insecurity being associated with poorer mental health [8,14].

Invalid conclusions about causality can occur when associations between variables established at the population level are assumed to arise from mechanisms

operating within individuals (the so-called ecological fallacy). Associations present between individuals may well be of different magnitude, and even different direction, to those present within individuals. This could occur if individuals with specific personality types are more likely to experience both food insecurity and poor mental health. The data summarised above come predominantly from cross-sectional studies and therefore provide no evidence for within-individual processes. Within-individual dynamics should be tested in longitudinal datasets in which it is possible to isolate within-individual covariance between fluctuations in food insecurity and fluctuations in mental health [15,16]. Longitudinal datasets additionally allow exploration of whether changes in food insecurity precede changes in mental health, as required for the third condition for causality. Granger causality is said to occur if a measure of past food insecurity improves the prediction of future mental health, compared with predictions based on past mental health alone [17].

A small number of longitudinal studies report crucial evidence for within-individual associations between fluctuations in food insecurity and poorer mental health [18–22]. Some find food insecurity predicts poorer mental health at a future time point [6,18,23,24] (see [25] for an exception), with a subset additionally providing evidence for a bidirectional relationship [26–28]. However, all existing studies have limitations, including: few data points per individual (two to four); long reference periods for the measurement of food insecurity and mental health (3 months to 1 year); and long intervals between data points (3 months to 2 years). These limit the ability to infer temporal patterns underlying observed associations if memory is poor or if effects weaken over time, leading to potentially misleading conclusions [29].

In the current study, we analysed covariation between measurements of food insecurity status and symptoms of anxiety and depression in two panels of adults from the UK (n = 244) and France (n = 240) collected as part of the Changing Cost of Living Study (CCLS; [30]). This intensive longitudinal study was conducted over 12 months between September 2022 and August 2023. Rates of food insecurity rose steeply in the UK in 2022. We therefore anticipated that some participants were likely to have experienced the fluctuations in food-insecurity status necessary to explore within-individual correlations with mental health. In the CCLS, food insecurity and mental health were measured monthly for an entire year, with reference periods of less than a month. Thus, individual participants have up to 12 consecutive data points, providing an unprecedented opportunity to explore short-term covariation between fluctuations in food insecurity and mental health.

The aims and analysis plan for the current study were preregistered on the Open Science Framework (OSF): https://osf.io/3rby2. Our objectives were as follows: (1) to describe month-to-month fluctuations in food insecurity and symptoms of depression and anxiety over the 12-month period; (2) to explore whether food-insecurity status predicted concurrent measures of symptoms of anxiety and depression, and establish the relative strength of the associations; (3) to isolate and test the effects of fluctuations in food-insecurity status within individuals on symptoms of anxiety and depression; and (4) to test whether the direction of changes in food-insecurity status affected the size of changes in symptoms of anxiety and depression. Finally, in a non-preregistered analysis to test for Granger causality, we explored whether adding food-insecurity status to statistical models improved the prediction of future symptoms of anxiety/depression compared with predictions based on past symptoms alone.

## Methods

### Ethics statement

The present study is a secondary analysis of an anonymised dataset that is in the public domain. The original Changing Cost of Living Study was approved by the Newcastle University Faculty of Medical Sciences Research Ethics Committee (approval number 2413/24908/2021). The data were accessed on 10th November 2023. The researchers had no access to information that could identify the original participants.

### Participants

The dataset was originally collected as part of the CCLS. Participant panels were recruited in UK (n = 244) and France (n = 240) via online research participation platforms. The panels were not nationally representative and were biased

towards the lower end of the income distribution, especially in France (see 30 for further details). Data were collected monthly, between September 2022 and August 2023. The full dataset and codebook are available at: https://osf.io/d9qb6.

## Measurement of food insecurity

Food insecurity was assessed with a reference period of the previous week using three items (*FI1*, *FI2* and *FI3*; Table 1). These items are the same as those in the UK Understanding Society Study [31] originally adapted from three items chosen from the eight-item UN Global Food Insecurity Experience Scale (FIES; [32]). Responses were summed to yield a single FI score (*FI*) ranging from 0 (food secure) to 3 (severe food insecurity). For the current analyses, we derived a dichotomous variable (*FI status*) by assigning participants as either food secure (*FI* = 0) or food insecure (*FI* > 0).

Since food insecurity varied both between and within individuals and our stated aim was to focus on within-individual variation, we created between- and within-individual versions of this variable. Following the methods in [15], we first made an integer version of *FI status* (food security coded 0 and food insecurity 1) and grand mean-centred each observation ($FI_c$). The resulting score was used to create a between-individual food-insecurity status score for each participant ($FI_{between}$ = mean($FI_c$)) and a within-individual food-insecurity status score for each observation ($FI_{within} = FI_c - FI_{between}$). $FI_{between}$ represents how food insecure a participant was on average over the year, compared to other participants. $FI_{within}$ represents whether, in a given month, a participant was more or less food insecure than their personal average level over the year.

## Mental health measures

Symptoms of anxiety were assessed with the seven-item Generalised Anxiety Disorder questionnaire (GAD-7). This was developed as a brief screening instrument for generalized anxiety disorder [33]. We used the continuous summary score (*GAD*) that ranged from 0 (no anxiety) to 21 (severe anxiety).

Symptoms of depression were assessed with the eight-item Patient Health Questionnaire (PHQ-8). This was developed as a brief screening instrument for depression based on eight of the core symptoms of depression in DSM-5 (PHQ-8; [34]). PHQ-8 is identical to PHQ-9 [35], but excludes the question about thoughts of death or hurting oneself. The PHQ-8 and PHQ-9 have been shown to be highly correlated (r = .997) and have similar sensitivity and specificity [34]. We used the continuous summary score (*PHQ*) that ranged from 0 (no depression) to 24 (severe depression).

For both scales, a score of 10 is regarded as the threshold for moderate to severe symptoms and clinically relevant disorder. Both GAD-7 and PHQ-8 were assessed with a reference period of the last two weeks. For statistical analysis, *GAD* and *PHQ* scores were square-root transformed due to positive skew.

**Table 1. Measurement of food insecurity.**

| Variable name | Question | Coding | Dimension assessed | Equivalent FIES item(s)* | FIES food-insecurity severity rating |
|---|---|---|---|---|---|
| FI1 | Thinking about last week, were you or others in your household unable to eat healthy and nutritious food because you could not afford it? | Yes (unable) = 1 No = 0 | Quality of food | Q2 | Mild |
| FI2 | Still thinking about last week, was there a time when you or others in your household were hungry but did not eat because you could not afford to? | Yes = 1 No = 0 | Hunger | Q7 | Severe |
| FI3 | Still thinking about last week, did you or other adults in the household have smaller meals than usual or skip meals because you could not afford or get access to food? | Yes = 1 No = 0 | Quantity of food | Q4 + Q5 | Moderate |

*FIES is the UN Global Food Insecurity Experience Scale [32].

## Statistical analysis

We used linear mixed-effect models (LMMs) in R (version 4.2.2) implemented in the package 'lme4'. An R script that reproduces the analyses and figures in this paper is available on the OSF: https://osf.io/3rby2. Model numbers correspond to those given in the preregistration. All models included a random intercept for participant, to account for non-independence due to repeated responses by the same participant. Random slopes for each participant were additionally included in preliminary models to check whether food insecurity and mental health scores changed in a consistent direction over the course of the study and in Models 5 and 6. The CCLS was not designed to study participants in the same household. While it is logically possible that some participants could have come from the same household, we have no means of knowing if this was the case and participants were therefore treated as independent in our analyses. We did not include the effect of country (UK or France) as either a fixed or random effect in any models, because the samples were not chosen to be nationally representative and the variance explained by country was negligible in previous analyses of this dataset [30]. In Models 1–3 we controlled for median age (median, because participants changed age during the study) and modal gender (mode, because some participants changed their reported gender identity during the study). Continuous predictor variables were grand-mean centred to facilitate interpretation of parameter estimates. Models used a Gaussian error structure where the outcome was anxiety or depressive symptoms. Where the outcome was food-insecurity status, we used a binomial error structure and logit link function. Where required, model fit was assessed using Akaike's Information Criterion (AIC). We assumed a criterion for statistical significance of $p < 0.05$.

Initial analyses of how our main variables varied over time (see Results for details) necessitated some changes to our preregistered analysis plan. First, due to the high intra-class correlation coefficients for GAD-7 and PHQ-8, in Models 1–6, we additionally controlled for mental health scores in the previous month to reduce the effect of autocorrelation in these variables. Second, because food insecurity, anxiety and depression all decreased linearly over the study, we also controlled for month (coded 0–11) to eliminate confounds due to time. In an additional departure, in Models 5 and 6 we used a method for isolating between- and within-individual effects of food insecurity that allowed us to model all the data points rather than using mean scores for each participant as originally proposed. These changes follow Bolger & Laurenceau's [15] recommendations for analysis of intensive longitudinal data. Table A in S1 Text provides details of all departures from the preregistered analysis plan.

## Results

### Descriptive statistics

Participant demographics are shown in Table 2, and descriptive statistics for the main measures reported each month are shown in Table 3. A total of 484 participants had food insecurity data for at least one of the 12 months of the CCLS and thus contribute to the current study (one participant in the original CCLS had no food insecurity data and was excluded). There was a strong positive correlation between GAD-7 and PHQ-8 scores, $r(4834) = 0.85$, $p < 0.001$.

### Objective 1: Month-to-month fluctuations in food insecurity and mental health

To compare repeatabilities of the main study variables within participants, we computed intra-class correlation coefficients (ICCs) for FI, GAD and PHQ scores. An ICC of 1 indicates that a given participant has the same value in every month, while an ICC of 0 indicates that participants vary from month to month as much as they differ from one another in a given month. The ICC was moderate for food-insecurity score (0.72 [95% CI: 0.69 - 0.74]), indicating variation both within and between individuals. ICCs were slightly higher for anxiety and depression scores (GAD: 0.79 [95% CI: 0.77 - 0.81]; PHQ: 0.8 [95% CI: 0.78 - 0.82]).

The monthly prevalence of food insecurity (i.e., an FI score $> 0$) averaged $15.87 \pm 2.33\%$ (M ± SD), and declined significantly over the study (Table B in S1 Text and Fig A in S1 Text). A total of 293 participants (60%) were food secure in all reported months, whereas 191 (39%) experienced at least one month of food insecurity, with 33 of these being food

**Table 2. Demographic descriptives of the sample.**

| | France (N = 240) | UK (N = 244) | Total (N = 484) |
|---|---|---|---|
| **Gender (N)** | | | |
| Man | 114 (47.5%) | 120 (49.2%) | 234 (48.3%) |
| Woman | 121 (50.4%) | 123 (50.4%) | 244 (50.4%) |
| PNTS or self-describe | 4 (1.7%) | 1 (0.4%) | 5 (1.0%) |
| Missing | 1 (0.4%) | 0 (0%) | 1 (0.2%) |
| **Age (years)** | | | |
| Mean (SD) | 41.2 (8.40) | 42.5 (12.1) | 41.9 (10.4) |
| **Children in household (N)** | | | |
| 0 | 136 (56.7%) | 152 (62.3%) | 288 (59.5%) |
| 1-2 | 92 (38.3%) | 86 (35.2%) | 178 (36.8%) |
| 3+ | 12 (5.0%) | 6 (2.5%) | 18 (3.7%) |
| **Overall food-insecurity status (N)** | | | |
| FS | 132 (55.0%) | 161 (66.0%) | 293 (60.5%) |
| FI | 108 (45.0%) | 83 (34.0%) | 191 (39.5%) |
| **Monthly reports available (number)** | | | |
| Mean (SD) | 9.27 (3.38) | 10.8 (2.32) | 10.0 (2.99) |

Ns in this table represent numbers of participants. Gender is the modal reported gender (PNTS: 'prefer not to say'), Age is the median reported age and Children is the modal reported number of children in the household. A participant was categorised as food-secure overall if their FI score was zero in all reported months and food-insecure if their FI score was greater than zero in any month.

**Table 3. Descriptive statistics for main study measures.**

| | France (N = 2238) | UK (N = 2642) | Total (N = 4880) |
|---|---|---|---|
| **Food insecurity (FI score)** | | | |
| Mean (SD) | 0.378 (0.844) | 0.232 (0.692) | 0.299 (0.769) |
| Missing | 14 (0.6%) | 4 (0.2%) | 18 (0.4%) |
| **Food-insecurity status (N)** | | | |
| FI | 451 (20.2%) | 327 (12.4%) | 778 (15.9%) |
| FS | 1773 (79.2%) | 2311 (87.5%) | 4084 (83.7%) |
| Missing | 14 (0.6%) | 4 (0.2%) | 18 (0.4%) |
| **Anxiety (GAD-7 score)** | | | |
| Mean (SD) | 5.72 (4.86) | 5.33 (5.76) | 5.50 (5.37) |
| Missing | 27 (1.2%) | 7 (0.3%) | 34 (0.7%) |
| **Depression (PHQ-8 score)** | | | |
| Mean (SD) | 6.25 (4.81) | 5.81 (6.13) | 6.01 (5.57) |
| Missing | 18 (0.8%) | 9 (0.3%) | 27 (0.6%) |

Ns in this table represent number of reports (up to 12 per participant).

insecure in all reported months. Therefore, 158 participants (33%) had at least one change in food-insecurity status over the course of the study. The percentage of participants that changed their food-insecurity status in a month averaged 9.3 ± 2.34% (M ± SD) and declined over the study (Fig A in S1 Text). Of the participants that changed their food-insecurity status, 11 experienced a single transition to food insecurity, 30 a single transition to food security and the remaining 117 (61 from the UK and 56 from France) had at least one transition in both directions. For participants that experienced a change in status, the average number of changes over the study was 2.61 ± 1.68 (M ± SD). Therefore, as required to

explore the within-individual effects, there were fluctuations in food-insecurity status over the course of the study within a substantial subset of participants.

Anxiety (GAD-7) and depression (PHQ-8) scores also declined significantly over the study (Table B and Fig A in S1 Text). GAD-7 scores were significantly higher in women than men (Table B in S1 Text).

### Objective 2: Associations between food insecurity and mental health

To test whether food-insecurity status predicted concurrent symptoms of anxiety and depression and compare the relative strength of associations, we fitted a single LMM (Model 1) with the continuous scaled GAD or PHQ score (*score*) as the outcome variable and scale type (*scale*: GAD versus PHQ), food-insecurity status (*FI status*: FS versus FI) and their interaction as fixed predictors. GAD and PHQ scores were standardized to have the same mean and standard deviation for this analysis to allow direct comparison of effects of food insecurity on symptoms of anxiety and depression. We additionally included fixed effects of gender and age to control for known associations between these variables and mental health. There was a significant main effect of *FI status*, with food-insecure participants having higher scores (Table 4; Fig 1A). The interaction between scale and food-insecurity status was not significant, indicating no evidence for differential associations of food insecurity with symptoms of anxiety and depression. Women had significantly higher GAD/PHQ scores than men and older participants had significantly lower scores. Scores declined significantly with month and were predicted by a participant's GAD/PHQ score in the previous month, despite controlling for month. Separate follow-up models with GAD and PHQ as the outcome variables (Models 2 and 3 respectively) confirmed that all these results held for both anxiety and depression separately (Table 4).

### Objective 3: Effects of fluctuations in food-insecurity status within individuals

To test within-individual effects of food insecurity on mental health, we used the subset of 158 participants who experienced fluctuation in their food-insecurity status during the study and split *FI status* into two orthogonal components: a between-individual means component, $FI_{between}$, and a within-individual deviations from those means component, $FI_{within}$ (see Methods for details). We fitted LMMs with the continuous scaled GAD or PHQ score as the outcome variable and $FI_{between}$ and $FI_{within}$ as predictors (Models 5 and 6). We allowed for variation between individuals in the response to food insecurity by including random slopes for the within-individual effect of food insecurity.

**Table 4. Output of models predicting scaled anxiety and depression scores from food-insecurity status.**

| Predictors | Model 1: Combined score | | | Model 2: GAD-7 | | | Model 3: PHQ-8 | | |
|---|---|---|---|---|---|---|---|---|---|
| | *Estimates* | *CI* | *p* | *Estimates* | *CI* | *p* | *Estimates* | *CI* | *p* |
| (Intercept) | -0.10 | -0.17 – -0.03 | **0.007** | -0.13 | -0.21 – -0.05 | **0.002** | -0.09 | -0.17 – -0.01 | **0.026** |
| FI status: FI | 0.16 | 0.10 – 0.21 | **<0.001** | 0.21 | 0.15 – 0.27 | **<0.001** | 0.17 | 0.11 – 0.23 | **<0.001** |
| Scale: PHQ | -0.01 | -0.03 – 0.01 | 0.564 | | | | | | |
| Age | -0.01 | -0.02 – -0.01 | **<0.001** | -0.02 | -0.02 – -0.01 | **<0.001** | -0.01 | -0.02 – -0.01 | **<0.001** |
| Gender: Woman | 0.14 | 0.04 – 0.24 | **0.007** | 0.18 | 0.07 – 0.30 | **0.002** | 0.12 | 0.01 – 0.23 | **0.036** |
| Gender: PNTS or self-describe | 0.37 | -0.27 – 1.01 | 0.251 | 0.28 | -0.46 – 1.02 | 0.466 | 0.55 | -0.17 – 1.27 | 0.137 |
| Month | -0.01 | -0.01 – -0.01 | **<0.001** | -0.01 | -0.01 – -0.00 | **0.002** | -0.01 | -0.02 – -0.01 | **<0.001** |
| Lagged score | 0.33 | 0.31 – 0.35 | **<0.001** | | | | | | |
| FI statusFI:scalePHQ | 0.02 | -0.04 – 0.07 | 0.527 | | | | | | |
| Lagged GAD | | | | 0.24 | 0.21 – 0.27 | **<0.001** | | | |
| Lagged PHQ | | | | | | | 0.27 | 0.24 – 0.29 | **<0.001** |
| Participants | 481 | | | 481 | | | 480 | | |
| Observations | 8710 | | | 4347 | | | 4363 | | |

PLOS Mental Health

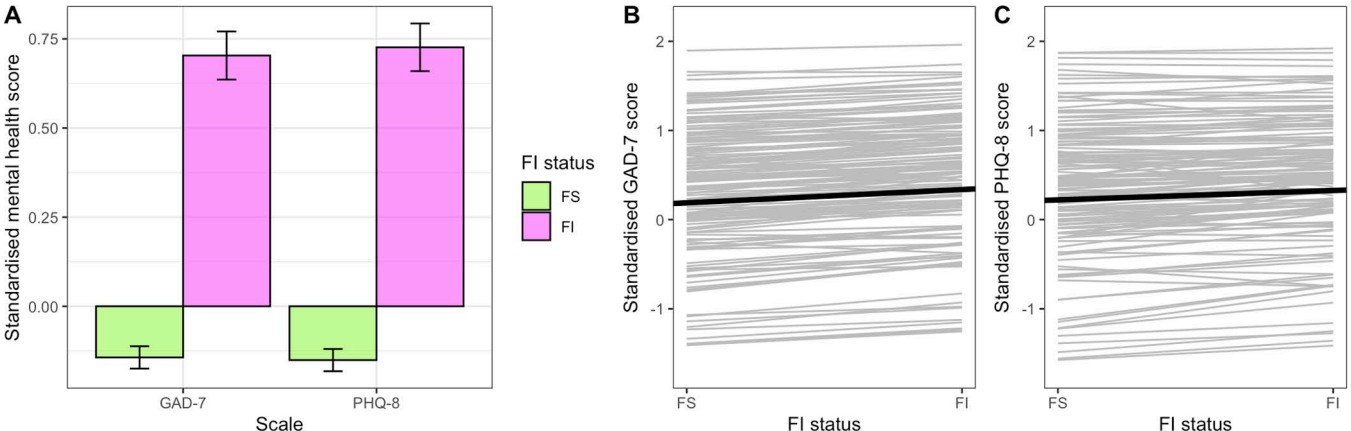

**Fig 1. Effects of food-insecurity status on symptoms of anxiety and depression. (A)** Combined within- and between-individual effects: bar graph showing mean standardised anxiety (GAD-7) and depression (PHQ-8) scores as a function of food-insecurity status. Data are monthly scores (N = 8710 observations from 481 participants). Error bars represent 95% confidence intervals. **(B and C)** Within-individual effects: spaghetti plots of average (thick black) and individual-specific (thin grey) fitted regression lines for (B) standardised GAD-7 score as a function of food-insecurity status (based on 1453 observations from 157 participants), and (C) standardised PHQ-8 score as a function of FI status (based on 1459 observations from 157 participants). Panels B and C are based on the subset of participants who changed their food-insecurity status at least once during the study.

Between individuals, there were large main effects of food-insecurity status on symptoms of both anxiety and depression, with participants who had higher mean food insecurity over the study also having higher mean GAD-7 and PHQ-8 scores. Additionally, there were small, within-individual effects: participants had higher GAD-7 and PHQ-8 scores in months when they were food insecure compared with months when they were food secure (Table 5; Figs 1B and 1C). The between--individual effects were 4.76 and 7.23 times the size of the within-individual effects for GAD-7 and PHQ-8 respectively.

### Objective 4: Effects of the direction of changes in food-insecurity status

To explore the effect of the direction of transitions in food-insecurity status on changes in mental health, we computed the change in GAD and PHQ scores at each transition for the subset of participants who experienced at

**Table 5. Output of models predicting mean standardised anxiety and depression scores from food-insecurity status in the subset of participants who changed their status during the study.**

| | Model 5: GAD | | | Model 6: PHQ | | |
|---|---|---|---|---|---|---|
| *Predictors* | *Estimates* | *CI* | *p* | *Estimates* | *CI* | *p* |
| (Intercept) | 0.19 | 0.05 – 0.33 | **0.009** | 0.22 | 0.09 – 0.35 | **0.001** |
| FI$_{within}$ | 0.15 | 0.08 – 0.22 | **<0.001** | 0.11 | 0.03 – 0.18 | **0.005** |
| FI$_{between}$ | 0.71 | 0.34 – 1.07 | **<0.001** | 0.76 | 0.42 – 1.11 | **<0.001** |
| Gender [Woman] | 0.15 | -0.03 – 0.34 | 0.110 | 0.05 | -0.13 – 0.23 | 0.592 |
| Gender [PNTS or self-describe] | 1.10 | -0.05 – 2.26 | 0.061 | 1.22 | 0.13 – 2.30 | **0.028** |
| Age | -0.02 | -0.03 – -0.01 | **0.002** | -0.01 | -0.02 – -0.01 | **0.002** |
| Month | -0.01 | -0.02 – 0.00 | 0.070 | -0.01 | -0.02 – -0.00 | **0.014** |
| Lagged GAD | 0.17 | 0.12 – 0.22 | **<0.001** | | | |
| Lagged PHQ | | | | 0.22 | 0.17 – 0.27 | **<0.001** |
| Participants | 158 | | | 158 | | |
| Observations | 1463 | | | 1469 | | |

least one transition in their food-insecurity status. We computed changes such that a deterioration in mental health accompanying a transition to food insecurity and an improvement in mental health accompanying a transition to food security would be scored in the same direction: $\Delta GAD = GAD_{FS} - GAD_{FI}$ and $\Delta PHQ = PHQ_{FS} - PHQ_{FI}$, where the subscripts indicate the food insecurity status at the time of a measurement. Thus, for both $\Delta GAD$ and $\Delta PHQ$, negative values always indicated poorer mental health scores under food insecurity. We additionally created a new variable, *direction,* that specified whether a transition was from food insecure to secure (*to FS*) or food secure to insecure (*to FI*). We fitted two LMMs with $\Delta GAD$ and $\Delta PHQ$ as the outcome variables and *direction* as a fixed predictor (Models 7 and 8). In both models, the intercept was negative (significantly so for $\Delta GAD$ and marginally non-significantly so for $\Delta PHQ$), indicating either the predicted increase in symptoms of anxiety and depression when participants transitioned to food insecurity, or the predicted decrease in symptoms when participants transitioned to food security. However, there was no evidence that the absolute size of this effect was affected by the direction of the transition (Table 6 and Fig 2).

### Exploratory tests for Granger causality

Granger causality is informally defined as occurring when a cause happens prior to an outcome and uniquely predicts variation in the outcome [17]. To test whether food insecurity in one month Granger caused increased symptoms of anxiety and depression in the following month, we compared the fit of 'univariate' LMMs, in which the mental health score was predicted by the mental health score in the preceding calendar month, with 'bivariate' LMMs, in which food-insecurity status in the previous calendar month was added as an additional predictor variable. Following the informal definition provided above, Granger causality is demonstrated if the fit of the bivariate model is better than that of the univariate model.

For GAD-7, the bivariate model fitted significantly better than the univariate model ($\Delta AIC = 6.02$, $\chi^2(1) = 8.02$, $p = 0.005$; Table C in S1 Text), indicating that food insecurity Granger causes increased symptoms of anxiety. For PHQ-8, the bivariate model also fitted better than the univariate model, but the difference in fit was marginally non-significant ($\Delta AIC = 0.89$, $\chi^2(1) = 2.89$, $p = 0.089$; Table D in S1 Text), indicating weaker evidence for Granger causality in the case of symptoms of depression.

### The impact of eliminating food insecurity on clinically relevant levels of anxiety and depression symptoms

To illustrate the likely impact of eliminating food insecurity on the prevalence of clinically relevant levels of anxiety and depression symptoms, we used the subset of monthly reports from the participants who changed their food-insecurity status at least once during the study. We chose this subset to emphasise within-individual effects (the effects are substantially larger if the full dataset is used). We divided the monthly reports into two groups according to whether the participant was categorised as food-secure or food-insecure in the month of the report. The observed distributions of GAD-7 and PHQ-8 scores in these groups are illustrated in Fig 3. In food-insecure months, 46.83% of GAD-7 scores and 47.02% of

**Table 6. Output of models predicting change in anxiety and depression scores from the direction of change of food-insecurity status within participants.**

| | Model 7: ΔGAD | | | Model 8: ΔPHQ | | |
|---|---|---|---|---|---|---|
| *Predictors* | *Estimates* | *CI* | *p* | *Estimates* | *CI* | *p* |
| (Intercept) | -0.71 | -1.28 – -0.14 | **0.014** | -0.53 | -1.15 – 0.09 | 0.095 |
| Direction [to FS] | -0.10 | -0.76 – 0.56 | 0.771 | -0.15 | -0.90 – 0.60 | 0.693 |
| Participants | 158 | | | 158 | | |
| Observations | 407 | | | 412 | | |

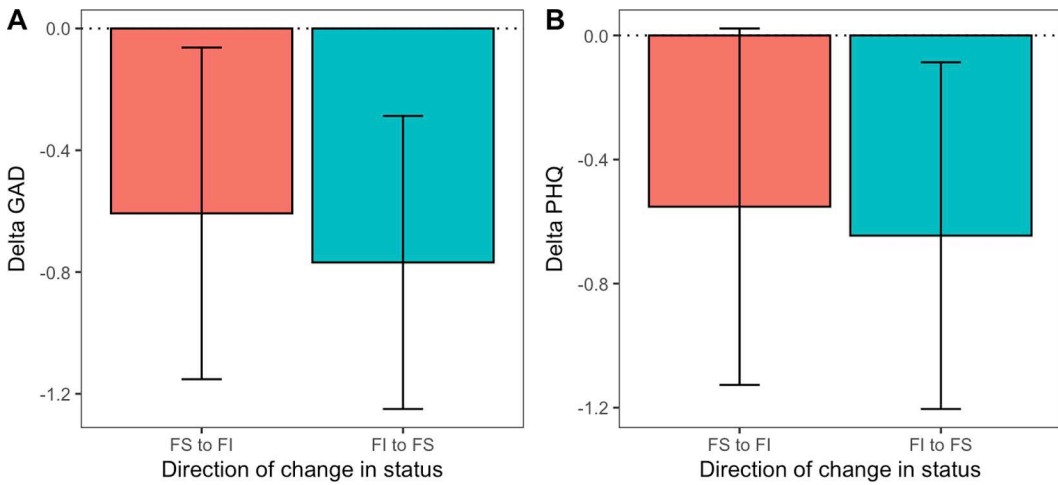

**Fig 2. Effect of direction of change in food-insecurity status on mental health:** (A) mean ΔGAD and (B) mean ΔPHQ. The negative delta values indicate that mental health measures were worse under food insecurity than food security. Error bars represent 95% confidence intervals.

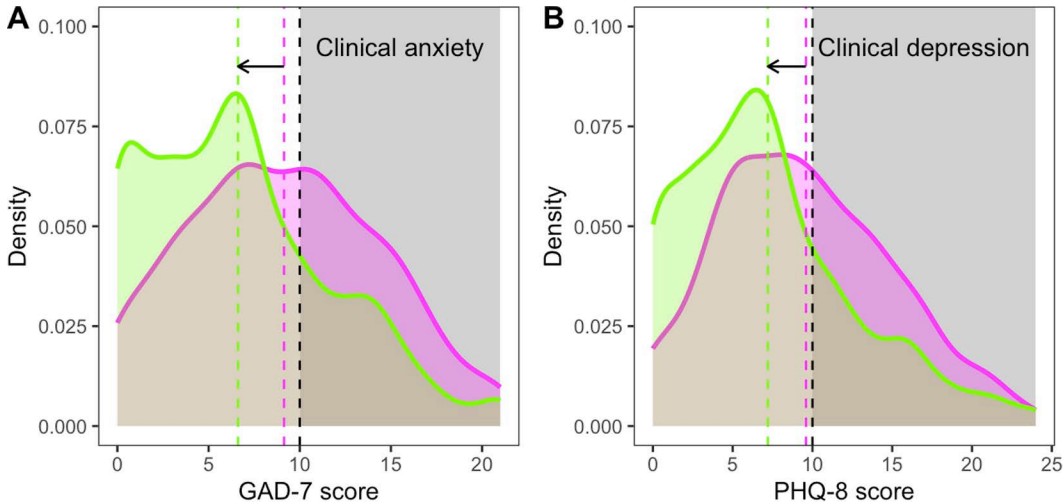

**Fig 3. Observed distributions of:** (A) GAD-7 scores, and (B) PHQ-8 scores in the subset of 158 participants whose food-insecurity status changed at least once during the study. The magenta curves show the distribution of scores in food-insecure months and the green curves the distribution in food-secure months. The magenta and green dashed lines show the mean scores for the food-insecure and food-secure months respectively and the arrow indicates the predicted shift in the mean score if food insecurity were eliminated in this sample. The black dashed lines show the threshold scores for clinical relevance.

PHQ-9 scores were greater than or equal to 10—the threshold for clinical concern on both scales. In food-secure months, these figures dropped to 26.04% for GAD-7 and 27.55% for PHQ-8. These data therefore suggest that eliminating food insecurity in individuals experiencing it periodically could potentially reduce the prevalence of clinically-relevant levels of anxiety and depression symptoms by 20.79 and 19.47 percentage points respectively. Note that due to concentrating only on the subset of participants who changed food-insecurity status during the study, this a conservative estimate that sets a lower bound on the potential impact of eliminating food insecurity.

## Discussion

To explore whether experience of food insecurity causes a deterioration in mental health, we analysed covariation between monthly measurements of food insecurity and symptoms of anxiety and depression using a panel dataset of adults from the UK and France. There was month-to-month variation in food-insecurity status and anxiety and depression symptoms over the 12-months of the study that allowed us to explore both between- and within-individual effects of food insecurity on mental health. Between individuals, those participants who were on average more food insecure during the study also on average had more symptoms of anxiety and depression. Within the subset of participants whose food-insecurity status fluctuated during the study, food-insecure months were associated with more symptoms of both anxiety and depression than the participant's personal averages for the study. In participants experiencing fluctuations in food-insecurity status, the deterioration in mental health when they became food insecure was of similar absolute magnitude to the improvement observed when they became food secure. Predictions of future mental health were better if they were based on the combination of mental health and food-insecurity status in the previous month, than if they were based on mental health in the previous month alone. We discuss each of these results below. We conclude that together they support the hypothesis that becoming food insecure causes a decline in mental health within one month, but that the effects are equally rapidly reversed when food security is reinstated.

### Prevalence of food insecurity

We observed a high overall prevalence of food insecurity, with 39.5% of participants experiencing food insecurity in at least one month of the study (34.0% in the UK sample). For comparison, in September 2022 (the beginning of CCLS data collection), the Food Foundation surveyed 4280 adults in the UK and reported 18.4% with food insecurity in the past month (https://foodfoundation.org.uk/initiatives/food-insecurity-tracking#tabs/Round-14). There are three possible reasons for the higher prevalence of food insecurity in the CCLS. First, our sample contained an over-representation of poorer participants who were more likely to experience food insecurity [30]. Second, the survey we used assessed mild, moderate and severe food insecurity, whereas the Food Foundation survey only assessed moderate to severe food insecurity. Finally, the CCLS used a non-typically short, 1-week reference period for the assessment of food insecurity (the Food Foundation used a 30-day reference period). We speculate that the short reference period used in the CCLS might have improved the quality of recall of milder hardships and hence increased the number of participants reporting some level of food insecurity at least once during the year of the study.

### Month-to-month fluctuations in food-insecurity status

We observed a trend of decreasing prevalence of food insecurity over the year, which likely reflected an average improvement in participants' financial situations over the same period [30]. A similar trend was reported by the Food Foundation in the UK, with the percentage of households experiencing moderate to severe food insecurity falling from 18.4% in September 2022 to 14.8% in January 2024.

Despite this consistent trend over time, the experience of individuals was highly dynamic, with the majority of food-insecure participants experiencing changes from both food insecurity to food security and vice versa. These month-to-month fluctuations are likely to have been accentuated by the short reference period used for assessing food insecurity, due to intermittent hardships not occurring during the week-long reference period for monthly reports. This latter possibility is supported by a comparison of results obtained from a food-insecurity survey conducted with long (1-year) and short (30-day) reference periods, which showed that the hardships described in survey items were more likely to be reported in the last year than in the last 30 days [36]. The number of fluctuations in food-insecurity status that we observed in the CCLS provided a unique opportunity, not present in studies with longer measurement reference periods, to explore the short-term effects of food insecurity on mental health.

## Associations between food-insecurity status and anxiety and depression symptoms

We found associations between food insecurity and concurrent measurements of symptoms of anxiety and depression (Models 1–3; Table 4 and Fig 1A). Being classified as food insecure was associated with having more symptoms of anxiety and depression. These associations were robust to controlling for potential confounds, including gender, age, time and mental health in the previous month. These results support the findings of many previous studies showing that food insecurity is associated with increased odds of both anxiety and depression. For example, a meta-analysis of data from 187,041 adults in the USA showed that the odds of depression were 2.74 (95% CI 2.52–2.97) times higher and those of anxiety were 2.41 (95% CI 1.81–3.22) times higher in food-insecure participants compared with food-secure participants [3]. Similarly, a meta-analysis of data from 372,143 adults in 19 countries showed that the odds of clinical depression were 1.40 (95% CI 1.30–1.58) times higher, and those of stress were 1.34 (95% CI 1.24–1.44) times higher in food-insecure compared with food-secure participants; the effect on anxiety while positive, was not significant (OR = 1.22; 95% CI: 0.98, 1.52) [5].

We found no evidence for a difference in the strength of the association between food insecurity and symptoms of anxiety and between food insecurity and symptoms of depression. The meta-analyses summarised above both report weaker effects of food insecurity on symptoms of anxiety than on symptoms of depression (although neither compares these effects directly, nor discusses the difference). Had this difference been supported, it could have had implications for the underlying mechanisms involved. To our knowledge, no previous study has directly compared associations between food insecurity and symptoms of anxiety and depression in the same individuals. To do this correctly, it is necessary to test the effect of the interaction between food insecurity and the mental health scale on outcome scores, as we did in Model 1. The lack of difference in effects on symptoms of anxiety and depression is not, on the face of it, surprising given the high correlation between GAD-7 and PHQ-8 scores in our dataset. Symptoms of anxiety and depression overlap and there is high comorbidity between these disorders [37].

A notable feature of our study is that the three items we used to assess food insecurity addressed the quality and quantity of food eaten and the experience of hunger, but not anxiety over future access to food (see Table 1). In contrast, the longer surveys more commonly used to assess food insecurity (involving 8 or more items) include an item addressing anxiety over future access to food. For example, FIES item 1 is, 'You were worried you would run out of food because of a lack of money or other resources' [32]. Thus, studies assessing the association between food insecurity and anxiety based on data from these long-form surveys of food insecurity suffer from an endogeneity due to an item addressing anxiety also being part of the assessment of food insecurity [12,38]. The fact that our study does not suffer from this trivial explanation for an association strengthens the evidence that the association we report is not a measurement artefact.

## Effects of fluctuations in food-insecurity status within individuals

Within participants whose food-insecurity status fluctuated during the study, food-insecure months were associated with more symptoms of anxiety and depression than the participant's personal averages for the study. In intensive longitudinal designs, such as that used in the current study, each participant effectively acts as their own control, thereby eliminating non-causal associations due to unmeasured individual differences. In order to fully capitalise on this feature of the design and test whether fluctuations in food insecurity were associated with fluctuations in mental health within participants, we constructed orthogonal predictor variables for between- and within-individual effects of food-insecurity status [15]. Models 5 and 6 showed that the effect of food insecurity on mental health was significant and in the same direction both between and within individuals, supporting a causal effect of food insecurity on mental health.

The impact of within-individual fluctuations in food insecurity on both anxiety and depression scores was substantially smaller than the between-individual effect (see Table 5). This between-within difference is commonly observed in longitudinal datasets (e.g., [30,39]). There are several possible reasons for this difference. First, the various proposed mechanistic pathways via which food insecurity causes a deterioration in mental health are likely to operate over different

timescales. Food insecurity is likely to produce acute psychological stress, but effects on mental health that occur as a result of nutrient deficiencies are likely to take longer to manifest. Hence, the impact of food insecurity on mental health is likely to be larger, the longer food insecurity is sustained. Our model, assessing the impact of food insecurity in the present month, thus gives a lower bound on the total within-individual effect, rather than capturing all of it. Second, while the within-individual effect is likely to reflect only the acute causal effect of food insecurity on mental health, the between-participant effect might additionally reflect the reverse causal effects of mental health on food insecurity, which are likely to take longer to manifest if they are mediated by less immediate consequences, such as loss of employment. Finally, it is possible that the between-individual effect includes the effects of residual confounding due to failure to control for variables that influence both the likelihood of reporting or becoming food insecure, and of reporting or developing symptoms of anxiety and depression.

### Effect of direction of change in food-insecurity status

We found that the deterioration in mental health observed when participants became food insecure was of similar absolute magnitude to the improvement when they became food secure (Fig 2). While longitudinal designs permit the isolation of within-individual associations, a limitation is that the putative causal conditions are experienced in a temporal order, raising the question of order effects. Order effects can occur for a range of reasons. One possibility is carry-over, whereby the effect of food insecurity persists even when food security is reinstated. Another possibility is that the effect of food insecurity might be worse the first time it is experienced, but less bad on each subsequent exposure as the individual learns to cope with it. Our results suggest that asymmetrical carry-over effects do not occur in the current study, and that eliminating food insecurity in individuals who experience it periodically could have immediate, positive effects on mental health.

We estimated that cases of clinically relevant levels of anxiety and depression symptoms in this sample would be reduced by 20.79 and 19.47 percentage points respectively if periodic food insecurity were eliminated (Fig 3). These figures translate into numbers needed to treat (NNT) for food security to prevent one case of clinical anxiety or depression of 5 and 6 respectively. Although an imperfect comparison, an NNT of 5 has been calculated for antidepressant drugs preventing relapse of major depressive disorder [40].

### Granger causality

In addition to showing that food-insecurity status predicted a participant's concurrent mental health, we took advantage of the time series nature of the CCLS dataset to demonstrate that food-insecurity status also predicted future mental health in the following month. Given that a putative cause must temporally precede an effect, this demonstration is a necessary criterion for causality, also known as Granger causality [41]. While formal tests of Granger causality require long time series of continuous variables [17], we adopted an informal definition that could be tested using the short time series available in the CCLS. We compared the fit of a univariate model, in which mental health was predicted by mental health in the preceding calendar month, with a bivariate model, in which food insecurity in the previous calendar month was added as an additional predictor variable. Consistent with food insecurity Granger causing reduced mental health, the fit of the bivariate model was better than that of the univariate model for predicting symptoms of both anxiety and depression. These results strengthen the case for a true causal effect of food insecurity on mental health.

In the current study, we did not explore temporal lags of more than one month due to the limitations of the dataset available. Therefore, our results say nothing about the longer-term effects on mental health (beyond one month) of experiencing a transition in food-insecurity status. Studies with both larger numbers of participants and longer time series of monthly within-individual measurements of food insecurity and mental health are required to address questions about longer-term effects of food insecurity.

## Limitations

The panels in the CCLS were not nationally representative. Given that our main conclusions concern the impact of within-individual fluctuations in food insecurity, this non-representativeness is not a serious methodological concern. However, the data reported here should not be used to estimate the population prevalence of food insecurity in the UK or France, or to make national comparisons.

The measurements of food insecurity and mental health were based entirely on self- reports by participants. Correlations between food-insecurity and mental health ratings may therefore be inflated due to shared method variance [38]. However, as discussed earlier, this concern is mitigated by our use of a short-form food insecurity survey that did not include an item explicitly addressing anxiety. We also note that anxiety and depression are defined in terms of subjective experience, and so all approaches to measuring them must include some kind of subjective report.

Finally, we stress that even intensive longitudinal designs, such as that used in the current study, do not permit causal inferences comparable to those afforded by experimental studies. However, given that it is currently not clear whether true experimental manipulation of food insecurity is either practically or ethically feasible, an intensive longitudinal design that allows exploration of within-individual effects is arguably the best we can currently achieve.

## Conclusions

Using monthly measurements of food insecurity, anxiety and depression from a panel of adults in the UK and France, we have demonstrated: (1) that fluctuations in food insecurity within individuals were associated with increased symptoms of anxiety and depression, (2) that these associations were unlikely to be caused by a confounding third variable, and (3) that food insecurity experienced in the previous month predicted a decline in mental health in the following month. Together these findings strongly suggest that food insecurity causes acute negative effects on mental health. Our results therefore contribute to a growing literature documenting the social determinants of mental health (e.g., [30,42]). Our analyses suggest that interventions to prevent food insecurity should produce a substantial and immediate reduction in the prevalence of clinically-relevant levels of anxiety and depression symptoms in populations currently experiencing periodic food insecurity, supporting the case for non-pharmaceutical treatments [43].

## Supporting information

**S1 Text.** Departures from preregistration, supplementary tables, supplementary figure.
(PDF)

## Acknowledgments

We thank Sarah Heaps for advice on Granger causality.

## Author contributions

**Conceptualization:** Melissa Bateson, Coralie Chevallier, Matthew T. Johnson, Kate E. Pickett, Daniel Nettle.

**Data curation:** Daniel Nettle.

**Formal analysis:** Melissa Bateson.

**Funding acquisition:** Coralie Chevallier, Matthew T. Johnson, Kate E. Pickett.

**Investigation:** Melissa Bateson, Elliott A. Johnson.

**Methodology:** Coralie Chevallier.

**Project administration:** Melissa Bateson, Coralie Chevallier.

**Visualization:** Melissa Bateson.

**Writing – original draft:** Melissa Bateson.

**Writing – review & editing:** Melissa Bateson, Coralie Chevallier, Elliott A. Johnson, Matthew T. Johnson, Kate E. Pickett, Daniel Nettle.

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
