## [Decision Letter · Decision Letter 0]

PMEN-D-25-00089

Does food insecurity cause anxiety and depression? Evidence from the Changing Cost of Living Study

PLOS Mental Health

Dear Dr. Nettle,

Thank you for submitting your manuscript to PLOS Mental Health. After careful consideration, we feel that it has merit but does not fully meet PLOS Mental Health’s publication criteria as it currently stands. Therefore, we invite you to submit a revised version of the manuscript that addresses the points raised during the review process.

Please find below detailed comments from the reviewers and provide a point-by-point response to these in your re-submission. 

We look forward to receiving your revised manuscript.

Kind regards,

Claire L. Niedzwiedz

Academic Editor

PLOS Mental Health

Journal Requirements:

Additional Editor Comments (if provided):

Reviewers' comments:

Reviewer's Responses to Questions

Comments to the Author

1. Does this manuscript meet PLOS Mental Health’s publication criteria?

Reviewer #1: Yes

Reviewer #2: Yes

2. Has the statistical analysis been performed appropriately and rigorously?

Reviewer #1: I don't know

Reviewer #2: Yes

3. Have the authors made all data underlying the findings in their manuscript fully available (please refer to the Data Availability Statement at the start of the manuscript PDF file)?

Reviewer #1: Yes

Reviewer #2: Yes

4. Is the manuscript presented in an intelligible fashion and written in standard English?

Reviewer #1: Yes

Reviewer #2: Yes

Reviewer #1: The study by Bateson et al, entitled: “Does food insecurity cause anxiety and depression? Evidence from the Changing Cost of Living Study” used an intensive longitudinal dataset of two cohorts -French and British, and explored evidence for causal association between food insecurity and symptoms of anxiety and depression (measured with GAD-7 and PHQ-8, respectively). The team found that fluctuations in food insecurity within individuals were associated with worse mental health and that food insecurity experienced in the previous month predicted a decline in mental health in the following month, and vice versa.

This was an interesting and well conducted study, methodology and reasons behind the methods used were well explained easing the understanding of the approaches even for a non-statistical audience. The findings and the discussion are clear. The authors are very open with sharing their outputs and datasets which is appreciated. I only have several comments/suggestions for improvement.

Please see the attached document for details.

Reviewer #2: This is a scientifically sound and needed study that examines the casual relationship between food insecurity and anxiety/depression symptoms. More scientific evidence like this are indeed necessary. I have the following comments that might improve the study further:

Introduction

- The authors should note that anxiety and depression symptoms are different from anxiety and depression as a disorder. The authors should therefore briefly discuss this in the introduction and use the appropriate term throughout the manuscript (there is is a lack of consistency in this) to ensure that it is clear to readers especially clinicians who might find the study's findings useful.

- The above should also be taken into consideration in the discussion and conclusion.

Methodology

- From lines 162 - 163, could the authors provide a justification for focusing on within-individual variation and not between or both?

- In line 177 change to 'Anxiety symptoms...' as this is what the GAD questionnaire in this study accessed.

- PHQ is used to access the severity of depressive disorders in individuals with depressive disorder. I imagine it is used to access depressive symptoms (which is one of its uses) in this study. This should therefore be reflected in line 181 that should be 'Depressive symptoms...'.

Discussion

- In line 540 does the term 'future mental health' refer to the one month prediction? The authors should clarify this statement especially because the following descriptions states that not more than one month was explored.

Conclusion

- The authors should give examples of such effects in line 581.

- I don't quit agree with the authors position in line 581 - 583.

Anxiety and depression are responses due to varying and most often compounding factors including an individuals socioeconomic/environmental exposure, genetic and biological dysregulations. Treatment therefore are tailored based on individual needs and involves medications (drugs), psychotherapy, nutritional interventions, or a varying combination of these. Also, anxiety and depression can negatively impact physical and mental health and vice versa either directly or indirectly. These in most cases requires medications with drugs. Refer to the following studies for more insights: https://doi.org/10.3390/nu16040501,
https://doi.org/10.1183/09059180.00007813,
https://doi.org/10.1016/j.actpsy.2024.104285,
https://www.ncbi.nlm.nih.gov/books/NBK559078/#:~:text=Go%20to:-,Etiology,low%20levels%20of%20serotonin%20metabolites.

- The authors should discuss based on lines 579 - 580 if this prediction is possible beyond the one month time period. If yes, how, and if not, why.

- From lines 584 - 586, give examples of such interventions. You can refer to the following study for such interventions: https://doi.org/10.3390/nu16040501

**Do you want your identity to be public for this peer review?** For information about this choice, including consent withdrawal, please see our Privacy Policy

Reviewer #1: No

Reviewer #2: **Yes: ** Ovinuchi Ejiohuo

---

## [Editor Report · Decision Letter 1]

Does food insecurity cause anxiety and depression? Evidence from the Changing Cost of Living Study

PMEN-D-25-00089R1

Dear Dr Nettle,

We are pleased to inform you that your manuscript 'Does food insecurity cause anxiety and depression? Evidence from the Changing Cost of Living Study' has been provisionally accepted for publication in PLOS Mental Health.

Best regards,

Claire L. Niedzwiedz

Academic Editor

PLOS Mental Health